# The bloodstream form of *Trypanosoma brucei* displays non-canonical gluconeogenesis

**Julie Kovářová**[1]*, **Martin Moos**[2], **Michael P. Barrett**[3], **David Horn**[4], **Alena Zíková**[1]

**1** Institute of Parasitology, Biology Centre of the Czech Academy of Sciences, České Budějovice, Czech Republic, **2** Institute of Entomology, Biology Centre of the Czech Academy of Sciences, České Budějovice, Czech Republic, **3** Wellcome Centre for Integrative Parasitology, University of Glasgow, Glasgow, United Kingdom, **4** Wellcome Centre for Anti-Infectives Research, University of Dundee, Dundee, United Kingdom

* julie.kovarova@paru.cas.cz

**Data Availability Statement:** The data is available from the Figshare database. https://figshare.com/authors/Julie_Kovarova/17814896.

## Abstract

*Trypanosoma brucei* is a causative agent of the Human and Animal African Trypanosomiases. The mammalian stage parasites infect various tissues and organs including the bloodstream, central nervous system, skin, adipose tissue and lungs. They rely on ATP produced in glycolysis, consuming large amounts of glucose, which is readily available in the mammalian host. In addition to glucose, glycerol can also be used as a source of carbon and ATP and as a substrate for gluconeogenesis. However, the physiological relevance of glycerol-fed gluconeogenesis for the mammalian-infective life cycle forms remains elusive. To demonstrate its (in)dispensability, first we must identify the enzyme(s) of the pathway. Loss of the canonical gluconeogenic enzyme, fructose-1,6-bisphosphatase, does not abolish the process hence at least one other enzyme must participate in gluconeogenesis in trypanosomes. Using a combination of CRISPR/Cas9 gene editing and RNA interference, we generated mutants for four enzymes potentially capable of contributing to gluconeogenesis: fructose-1,6-bisphoshatase, sedoheptulose-1,7-bisphosphatase, phosphofructokinase and transaldolase, alone or in various combinations. Metabolomic analyses revealed that flux through gluconeogenesis was maintained irrespective of which of these genes were lost. Our data render unlikely a previously hypothesised role of a reverse phosphofructokinase reaction in gluconeogenesis and preclude the participation of a novel biochemical pathway involving transaldolase in the process. The sustained metabolic flux in gluconeogenesis in our mutants, including a triple-null strain, indicates the presence of a unique enzyme participating in gluconeogenesis. Additionally, the data provide new insights into gluconeogenesis and the pentose phosphate pathway, and improve the current understanding of carbon metabolism of the mammalian-infective stages of *T. brucei*.

## Author summary

*Trypanosoma brucei* is a unicellular parasite causing sleeping sickness in humans and nagana disease in cattle. The parasite invades the bloodstream and cerebrospinal fluid and only recently, it has been shown to infect additional tissues such as skin, adipose tissue, or

**Funding:** This work was supported by CZ.02.2.69/ 0.0/0.0/19_074/0016248 LeishWeb to J.K.; by the Ministry of Education, Youth and Sports of the Czech Republic grant RNA for therapy (CZ.02.01.01/00/22_008/0004575), by the European Research Council (ERC, MitoSignal, grant agreement no. 101044951), and GACR 20-14409S to A.Z.; D. H. was funded by an Investigator Award from Wellcome [217105/Z/19/ Z]. M.P.B was funded by an MRC Newton grant 'Bridging epigenetics, metabolism and cell cycle in pathogenic trypanosomatids' MR/S019650/1. The funders had no role in study design, data collection and analysis, decision to publish, or preparation of the manuscript.

**Competing interests:** The authors have declared that no competing interests exist.

lungs. While the glucose-based metabolism of the bloodstream form is well understood, the parasite's metabolism in these secondary tissues has not been sufficiently explored, despite its importance for drug development. One possibility is the use of gluconeogenesis since the mammalian-infective stages can use glycerol as a carbon and ATP source. First, enzymes involved in gluconeogenesis have to be identified, then it can be tested if the pathway is advantageous for the survival of the parasite. We generated mutants in four different enzymes potentially involved in this metabolic pathway. Surprisingly, the flux in gluconeogenesis was maintained in all cell lines tested, implying that another non-canonical enzyme participates in the production of glucose from glycerol in these parasites.

## Introduction

*Trypanosoma brucei brucei* is the causative agent of Human African Trypanosomiasis, also termed sleeping sickness [1]. The mammalian-infective stage of the parasite is called the bloodstream form (BSF), and it is transmitted between hosts by blood-feeding tsetse flies. In the first stage of the disease, these extracellular parasites divide in the bloodstream of the mammalian host. If left untreated, the trypanosomes invade the central nervous system, manifesting as the second stage of the disease. BSF parasites can also inhabit skin, adipose tissue, lungs and other tissues [1–5]. Previous dogma stated that BSF trypanosomes are absolutely glucose-dependent. However, we [6], and others [7] have shown that they also employ gluconeogenesis (GNG) and can use glycerol as a carbon source. Glycerol utilisation is expected to be most physiologically relevant to parasites that inhabit adipose tissue or skin, but the significance of glycerol as a carbon source remains elusive. Nevertheless, it is clear that the parasite's metabolism is highly flexible and adaptable, and may differ between the various mammalian forms, which should be considered when developing drugs inhibiting metabolic enzymes. Notably, the adipose tissue forms are less responsive to several trypanocidal drugs [8].

Under standard culture conditions in medium containing glucose, glycolysis provides the majority of cellular ATP and is indispensable to BSF trypanosomes [9]. However, in culture medium containing glycerol, BSF trypanosomes use GNG to produce sugars from non-sugar carbon sources by converting glycerol to glucose 6-phosphate (G6P) [6,7]. GNG primarily uses the same enzymes as glycolysis, but operating in the opposite direction. The key difference between the two pathways is the enzymatic step between fructose 6-phosphate (F6P), and fructose 1,6-bisphosphate (F1,6bP) when phosphofructokinase (PFK) phosphorylates F6P to F1,6bP in glycolysis, while fructose-1,6-bisphosphatase (FBPase) dephosphorylates F1,6bP to form F6P in GNG (Fig 1A) [10]. In addition, since glycerol is used as the non-sugar substrate, glycerol kinase (GK) becomes a key GNG enzyme [7].

Previously, we demonstrated the presence of GNG in the BSF parasites by RNA interference (RNAi) silencing of the glucose transporters, where the observed lethal phenotype was rescued by the addition of glycerol to the culture media [6]. As verified by LC-MS metabolomics with $^{13}$C-glycerol, glycerol was incorporated into fructose 6-phosphate (F6P) and other metabolites via GNG. The same conclusion was reached by Pineda and colleagues after adapting BSF parasites to glucose-free, glycerol-containing medium [7]. Surprisingly, however, GNG was not abolished after deletion of the *FBPase* gene indicating an involvement of another, so far unknown, enzyme [6,7].

The role of FBPase in GNG was also studied in the insect procyclic form (PCF) of *T. brucei* [11]. This stage has a more elaborate mitochondrion in terms of both morphology and metabolism, which can provide additional substrates for GNG. Hence, proline is utilised by PCF as a

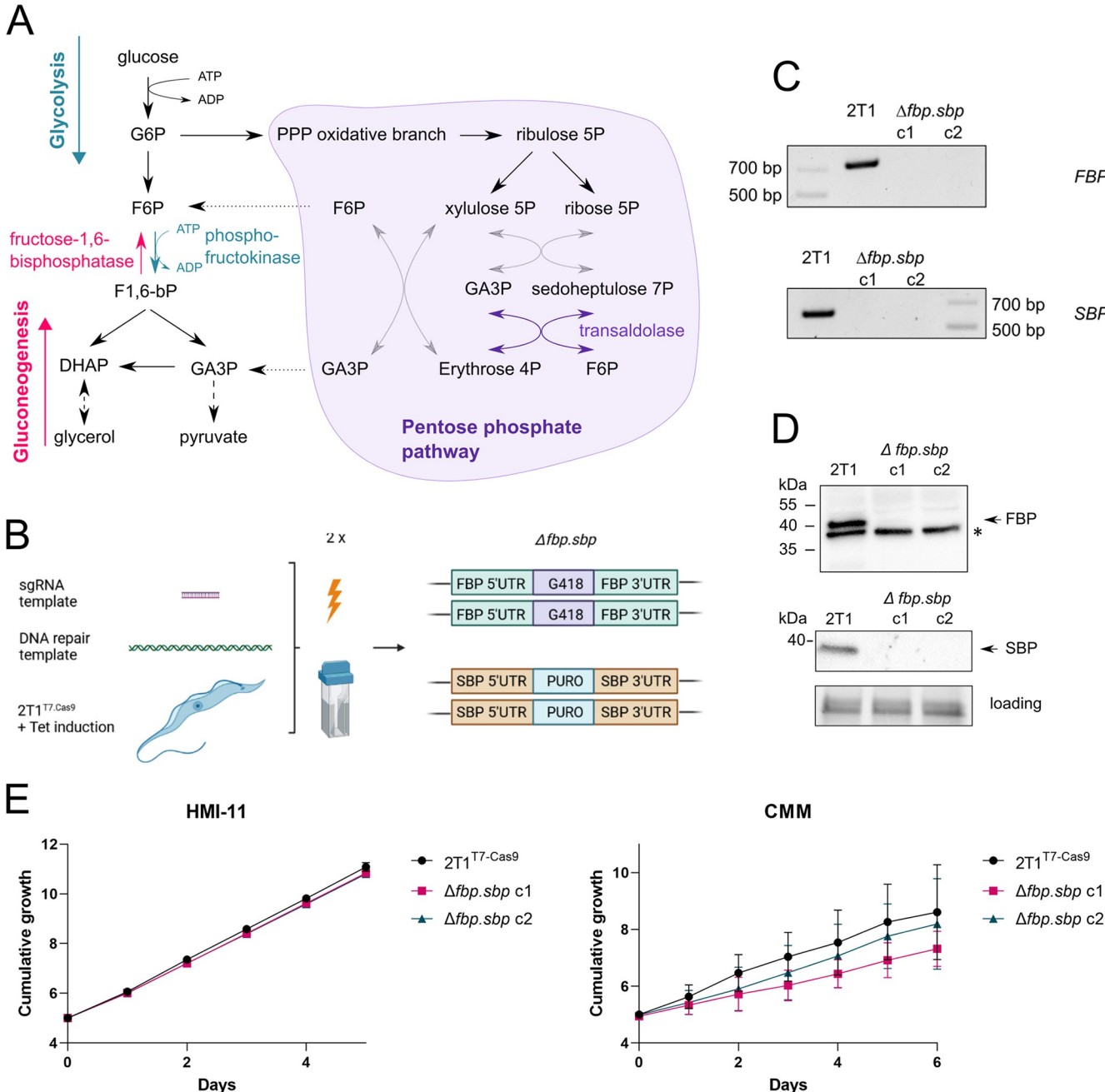

**Fig 1. Generation and growth analysis of Δ*fbp.sbp* strains.** A–The scheme shows glycolysis, gluconeogenesis, and the pentose phosphate pathway in BSF *T. brucei* with highlighted FBPase, PFK, and TAL. Missing reactions of transketolase are depicted by grey arrows. B–A scheme for the Cas9 editing method by transient transfection. The templates for an sgRNA and for an antibiotic resistance cassette were transfected simultaneously, which resulted in replacement of both alleles for *FBPase* in the first step, and for *SBPase* in the second step. Fig 1B was created in BioRender. C–Validation of the Δ*fbp.sbp* cell line by PCR, shows parts of ORFs for *FBPase* and *SBPase* amplified in the parental 2T1[T7.Cas9] cell line, but absent from Δ*fbp.sbp*. D—Validation of the Δ*fbp.sbp* cell line by western blot. Only the parental cell line shows signal for FBPase (whole cell lysates) and SBPase (organellar fractions), *—cross-reacting protein, gel loading–fluorescent protein detection on a TGX gel. E–Growth curves of two independent clones of Δ*fbp.sbp* show no defect in the standard HMI-11 medium. Growth curves in the CMM medium show a mild growth defect of the Δ*fbp.sbp* clones and higher variability.

carbon source, via proline degradation in the mitochondrion, and fed into GNG via phospho-enolpyruvate carboxykinase and pyruvate phosphate dikinase [11]. FBPase knock-out (KO) PCF cells are viable when grown in medium containing proline and they incorporate proline-derived metabolites into G6P by GNG. This suggests that, similarly to BSF cells, the activity of FBPase can be compensated by another unknown enzyme. Interestingly, the FBPase KO PCF cells display a mild growth defect, defects in metacyclogenesis and transmission through the tsetse fly [11].

The presence of another enzyme catalyzing the F1,6bP to F6P conversion is intriguing, as in the majority of eukaryotic cells studied to date, this metabolic reaction is performed solely by the activity of FBPase. Therefore, the balance between FBPase and PFK activity is strictly and tightly regulated [10]. FBPase is positively regulated by ATP, and negatively by AMP and fructose 2,6-bisphosphate (F2,6bP). PFK is controlled by the same metabolites in a reciprocal manner, i.e. it is positively regulated by AMP, ADP, and F2,6bP. F2,6bP is an activator of most eukaryotic PFK enzymes, normally produced from F1,6bP by the bifunctional enzyme, 6-phosphofructo-2-kinase/fructose-2,6-bisphosphatase [12]. The finely tuned regulatory mechanism between PFK and FBPase ensures mutual exclusivity of these two reactions, and prevents futile cycling (a situation when both reactions would be running in a cycle, only consuming ATP). Interestingly, *T. brucei* PFK is much less sensitive to inhibition by ATP than many eukaryotic PFKs, and its activity is not affected by the presence of F2,6bP [13]. Therefore, the well-established PFK regulatory mechanism appears to be missing from *T. brucei* [14]. Regulation of *T. brucei* FBPase has not yet been elucidated.

Although *T. brucei* PFK is an ATP-dependent enzyme, its amino acid sequence is a 'chimera' between representative pyrophosphate- and ATP-dependent enzymes [15]. PFK isolated from *T. brucei* is regulated by AMP, which serves as the only allosteric activator, although 10-fold less potent than for the leishmania enzyme [13,15], and by phosphoenolpyruvate acting as an allosteric inhibitor. F1,6bP or F2,6bP did not significantly influence enzyme activity [13]. As predicted by a mathematical model, and experimentally validated, PFK is present in excess in *T. brucei* and therefore, glycolytic flux is reduced only after depletion of PFK beyond 60% of WT activity. It is unclear, however, whether the effect is direct, or a secondary consequence of decreased activity of other glycolytic enzymes (hexokinase, enolase, pyruvate kinase) [9]. Highly specific allosteric inhibitors of *T. brucei* PFK have been developed, and shown to be effective in killing BSF parasites *in vitro* and in infected mice [16]. Fernandes and colleagues [17] reported reverse activity of trypanosomal and mammalian PFKs *in vitro*, but the physiological relevance is unclear. The reverse activity of three mammalian PFK isoforms is not thought to occur *in vivo* [18] and although trypanosomal PFK is localised in specialized organelles called glycosomes, which could theoretically form an environment that allows reverse activity, experimental evidence is still lacking.

In trypanosomatids, glycosomes are highly adapted peroxisomes, as they share related PEX-dependent protein import machineries. The main role of these specialized organelles is to harbor both glycolytic and gluconeogenic enzymes, parts of the pentose phosphate pathway (PPP), nucleotide metabolism, and other associated pathways [19]. The arrangement and function of the PPP is not fully resolved in *T.brucei* BSF because transketolase is not expressed in this stage, but transaldolase is, resulting in an incomplete non-oxidative branch of the PPP (Fig 1A) [20, 21]. The glycosomal membrane is 'semi-permeable', thought to allow free passage of small molecules (up to 340 Da based on a mathematical model of trypanosome metabolism) through open channels, but to be impermeable to larger molecules [22,23]. Hence, cofactors such as ADP/ATP or NAD$^+$/NADH are balanced inside glycosomes, and glycosomal localisation of the glycolytic enzymes has important implications for their regulation [24].

We and others have shown previously that deletion of the *FBPase* gene does not eliminate the enzymatic activity responsible for dephosphorylation of F1,6bP [6,7], indicating that other enzymes capable of this activity are also present. Notably, a gene encoding sedoheptulose-1,7-bisphosphatase (SBPase), an enzyme typically involved in the Calvin cycle of plants, is present in the genome of *T. brucei*. Due to a relatively high sequence identity (26%) to FBPase, we considered it a possible candidate for possessing FBPase activity. We also explored two other scenarios in which the reverse PFK activity [17] or a pathway involving transaldolase (TAL) [25] could bypass FBPase in GNG. To our surprise, double knock-out cell lines for FBPase and SBPase (*Δfbp.sbp*), triple knock out cell lines for FBPas, SBPase and TAL (*Δfbp.sbp.tal*) and PFK RNAi cell lines in a background of *Δfbp.sbp*, retained GNG flux, suggesting an unknown enzyme performing the FBPase reaction. Nevertheless, we present several unique metabolomic datasets that provide new insights into non-canonical GNG and PPP in BSF trypanosomes.

## Materials and methods

### *T. brucei* cell culture and cell line construction

Bloodstream form *Trypanosoma brucei brucei* Lister 427 cells were cultured in HMI-11 medium [26] supplemented with 10% FCS (BioSera) at 37˚C and 5% $CO_2$. The fructose-1,6-bisphosphatase and sedoheptulose-1,7-bisphosphatase knock-out cell line (*Δfbp.sbp*) was generated by CRISPR/Cas9 gene editing in the 2T1[T7-Cas9] cell line [27]. In order to delete *FBP* (Tb927.9.8720), a template for sgRNA transcription was synthesised by end-filling PCR with primers FBPgRNA_F and sgRNA_R, and a repair template for DNA integration was synthesised with primers FBP.NPT50_5 and FBP.NPT50_3. Cas9 expression was induced with tetracycline at 1 μg/ml (InvivoGen) 24 h prior to transfection with an Amaxa nucleofector (Lonza) using programme Z-001. First, the *Δfbp* cell line was generated, and after validation of successful replacement of both *FBP* alleles with a G418 resistance cassette, a subsequent transfection was performed to replace the *SBP* (Tb927.2.5800) alleles with a puromycin resistance cassette (primer pairs SBPgRNA_F and sgRNA_R for sgRNA, and SBP.PAC50_5 and SBP.PAC50_3 for the repair template with puromycin resistance). Likewise, *Δtal* (TAL is encoded by Tb927.8.5600) was generated in the parental and *Δfbp.sbp* cell lines, using primer pairs AZ1550 and sgRNA_R for sgRNA, and AZ1551 and AZ1552 for the repair template with phleomycin resistance. The primers used are listed in Table 1. Hygromycin was used at 5 μg/ml (InvivoGen), blasticidin at 5 μg/ml (InvivoGen), G418 at 2.5 μg/ml (Invivogen), and puromycin at 0.1 μg/ml (InvivoGen).

For depletion of PFK (Tb927.3.3270), the RNAi plasmid p2T7-177 [28] was used, digested with *Bam*HI and *Hind*III. A *PFK* PCR product was amplified using primer pair AZ1339 and AZ1340, digested with the same restriction enzymes and ligated with the plasmid. The resulting p2T7-177-PFK construct was linearized with *Not*I prior to electroporation of 2T1[T7-Cas9] and *Δfbp.sbp* cell lines in parallel. Clones obtained after phleomycin selection (at 2.5 μg/ml) were tested for a growth defect after tetracycline induction at 1 μg/ml, and validated by qRT-PCR.

The [Ty]PFK cell line was also created by Cas9 editing in the 2T1[T7-Cas9] cell line. Following 24 h of Cas9 induction, electroporation was performed with a sgRNA template (primer pair AZ1388 and sgRNA_R), and PURO.[Ty]PFK cassette (primer pair AZ1281 and AZ1282). Clones were selected with puromycin at 0.1 μg/ml and integration validated by PCR (primer pair AZ1403 and AZ1404) and western blot.

**Table 1. Sequences of DNA primers used in the study.**

| Primer | Sequence | Description |
|---|---|---|
| FBPgRNA_F | TAATACGACTCACTATAGGGGAGGGTGTGGTGCTTTTCGTGGGGCGTTTTAGAGCTAGAAATAGCAAG | FBP F sgRNA |
| sgRNA_R | GCACCGACTCGGTGCCACTTTTTCAAGTTGATAACGGACTAGCCTTATTTTAACTTGCTATTTCTAGCTCTAAAAC | R sgRNA universal |
| FBP.NPT50_5 | ttttgcgactgtttttcaatttcattaacgacaccactcttcccagatttATGATTGAACAAGATGGATTGC | FBP.G418 cassette F |
| FBP.NPT50_3 | TTTTAAAAAAAACCTGTACCCTTTCCACACGCATCGAAGCAACCATTGGCTCAGAAGAACTCGTCAAGAAGG | FBP.G418 cassette R |
| SBPgRNA_F | TAATACGACTCACTATAGGG TGTGGTGCTTTTCGTGGGGCGTTTTAGAGCTAGAAATAGCAAG | SBP F sgRNA |
| SBP.PAC50_5 | gcccgttacagggggttccctttcactcaacctttcgtggaaagggagaatATGACCGAGTACAAGCCC | SBP.puro cassette F |
| SBP.PAC50_3 | gatcggaccatcaccttgtgcggcacaaaaataaaaacaagaagaaaaaaGGTACCGAGCTCGAATTCTC | SBP.puro cassette R |
| AZ1211 | GATGGCGATTCAAACGTCCG | *FBP* ORF 93–812 bp |
| AZ1212 | TTATCCCCCGGGTAGCAGAA | *FBP* ORF 93–812 bp |
| AZ1213 | ATTGGCTTCTACCGCGCATA | *SBP* ORF 276–888 bp |
| AZ1214 | GTTCGTCAACCCCGTTGTTG | *SBP* ORF 276–888 bp |
| AZ1339 | ATGGATCCTTCACTTCAACCCGACGGAG | PFK RNAi F, *PFK* 338–894 bp |
| AZ1340 | ATAAGCTTCTGTCACGACCCATGAGCTT | PFK RNAi R, *PFK* 338–894 bp |
| AZ1388 | TAATACGACTCACTATAGGGGGTGACGAGCTTGCTCGTAACGGTTTTAGAGCTAGAAATAGCAAG | Ty.PFK sgRNA |
| AZ1281 | GTGAACTGGAGGCAATCAACAGGAATCGCGACCGCCTCCACGAGGAACTGGAGGTCCA TACTAACCAAGATCCACTTGACGCCAAGCTCTAATCTCCGCTCTTATTTAGTTTTGC | PURO.^TyPFK cassette F |
| AZ1282 | GTGCCCTCCTCATTCTCCCTTTCCCAGAACTCCGTTACCAGAGGGTCATTCAATCATGTCGACACACCAAG | PURO.^TyPFK cassette R |
| AZ1403 | GGACGTATTGCATGTGCTGTC | validation of ^TyPFK F |
| AZ1404 | TGAGTCACGCTGTTCAGCAT | validation of ^TyPFK R |
| AZ1550 | TAATACGACTCACTATAGGGCGCTACACTTTTGTAAGGGAGTTTTAGAGCTAGAAATAGCAAG | TAL sgRNA |
| AZ1551 | CTTAGAAGGGGGAAGGCAACAAGACATGGCCAAGTTGACCAGTG | TAL phleo cassette F |
| AZ1552 | GTTACTTGATGGTGGGTACCCTCCCTCAGTCCTGCTCCTCGG | TAL phleo cassette R |
| AZ1609 | ATGGATCCGTCTCTCGGCTACGTGATGG | validation of *Δtal* |
| AZ1610 | ATAAGCTTTCGTTACAGCAATGACGCCT | validation of *Δtal* |
| AZ1557 | GCTTAGAAGGGGGAAGGCAA | amplification of *TAL* locus F |
| AZ1558 | CGGGGAACGGTACTGTCAAA | amplification of *TAL* locus R |

## qRT-PCR

Total RNA was extracted from 1–2 x $10^8$ cells using the RNeasy kit (Qiagen). DNA was removed using Turbo DNase (Applichem) at 37˚ C for 30 min, which was subsequently treated with DNase inactivation reagent (Ambion) for 5 min at RT. Following ethanol precipitation, cDNA was synthesised from 2 μg of RNA using TaqMan Reverse Transcription Reagent (Applied Biosystems) and random hexamer primers. Real-time PCR amplification was performed using LightCycler 480 SYBR Green I Master (Roche) and LightCycler 480 thermocycler (Roche). Primers used for the PFK target and reference genes are listed in Table 2.

## Subcellular fractionation and western blotting

In order to separate organellar fractions containing glycosomes from the cytosol, cells were subjected to digitonin-based subcellular fractionation and the obtained samples used for western blots. Briefly, 1 x $10^8$ cells were harvested, washed in 1 x PBS and resuspended in 500 μl of

**Table 2. Sequences of DNA primers used for qRT-PCR.**

| Target gene | Primer name | Sequence |
|---|---|---|
| PFK F | AZ1489 | CCTCACGGAGAAAGTGAAGG |
| PFK R | AZ1490 | GGGTAGCGAGACTTGTTTGC |
| 18S F | AZ50 | GCGAAACGCCAAGCTAATAC |
| 18S R | AZ51 | AGCCGCGACATAGAAAAAGA |
| Tubulin F | AZ52 | GCAGAGTCCAACATGAACGA |
| Tubulin R | AZ53 | CGTCCGCGTCTAGTATTGCT |

SoTE buffer (0.6 M sorbitol, 2 mM EDTA, 20 mM Tris-HCl, pH 7.5). A further 500 μl of SoTE buffer containing 0.03% digitonin (Sigma-Aldrich) was added, samples were incubated on ice for 5 min and subsequently centrifuged at 4,500 g, 4˚ C for 3 min. The obtained supernatant was used as cytosolic fraction, and pellets resuspended in an equivalent volume of 1 x PBS and used as organellar fractions. 40 μl of samples were loaded for western blots.

For whole cell lysates, the equivalent of $1 \times 10^7$ cells was used. For western blots (WB) 4–12% NuPAGE polyacrylamide gels (Invitrogen) and 1 x SDS running buffer (25 mM Tris, 192 mM glycine, 1% SDS) were used. Subsequently, proteins were transferred to a PVDF membrane (Pierce) in transfer buffer (39 mM glycine, 48 mM Tris, 20% methanol) at 90 V for 90 min at 4˚C. Following 30 min blocking in 5% milk (Serva) in PBS-Tween (0.05%), primary antibody was incubated in milk solution overnight at 4˚C. Following 3 x 10 min wash in PBS-Tween, secondary antibody was incubated in milk solution for 1 h at RT. Following 3 x wash in PBS-T, signal was visualised using Western ECL Substrate (BioRad). The following antibodies were used: α-FBP at 1:500, α-SBP at 1:500 (both kind gifts from Frédéric Bringaud), anti-Ty 1:1,000 (ThermoFisher), α-APRT at 1:500 was used as a marker for cytosolic fraction, and α-hexokinase at 1:2,000 as an organellar marker. Secondary α-mouse (BioRad) and α-rabbit (BioRad) antibodies conjugated to HRP were used at 1:2,000.

## Metabolomics

For the experiment with *Δfbp.sbp* in $^{13}$C-glycerol, cells were grown in the standard HMI-11 medium supplemented with 5 mM $^{13}$C-U-glycerol (Cambridge Isotope Laboratories). For the experiment with *Δfbp.sbp*/$^{RNAi}$PFK, *Δtal*, and *Δfbp.sbp.tal*, HMI-11 medium was prepared from components according to the recipe [26], but glucose was omitted and instead 5 mM $^{13}$C$_3$-U-glycerol (Cambridge Isotope Laboratories) was supplied. Cells were grown in this medium for 24 h (PFK cell lines) or 48 h (TAL cell lines) prior to sample extraction.

Samples for the metabolomic experiments were prepared by the same extraction protocol as reported previously [29]. Briefly, $5 \times 10^7$ cells were used per 100 μl sample, which were first rapidly cooled to 4˚ C in a dry ice–ethanol bath. Following a wash with 1 x PBS, cell pellets were resuspended in 100 μl of chloroform:methanol:water (1:3:1) suspension and incubated with shaking at 4˚ C for 1 h in order to achieve full extraction into the solvent. Subsequently, samples were centrifuged (12,000 g, 10 min, 4˚ C), supernatants were collected and stored at -80˚ C until analysis.

The metabolomic methods used were described in detail elswere [30]. Briefly, an Orbitrap Q Exactive Plus mass spectrometer coupled to an LC Dionex Ultimate 3000 (Thermo Fisher Scientific, San Jose, CA, USA) was used for metabolite profiling. LC condition: column SeQuant ZIC-pHILIC 150 mm x 4.6 mm i.d., 5 μm, (Merck KGaA, Darmstadt, Germany); flow rate of 450 μl/min; injection volume of 5 μl; column temperature of 35˚C; mobile phase A = acetonitrile and B = 20 mmol/l aqueous ammonium carbonate (pH = 9.2; adjusted with

NH$_4$OH); gradient: 0 min, 20% B; 20 min, 80% B; 20.1 min, 95% B; 23.3 min, 95% B; 23.4 min, 20% B; 30.0 min 20% B. The Q-Exactive settings were: mass range 70–1050 Daltons; 70 000 resolution; electrospray ion source operated in the positive and negative modes.

The analysis of the *Δfbp.sbp* cell line in HMI-11 medium was performed at Glasgow Polyomics, using separation on 150 x 4.6 mm ZIC-pHILIC (Merck) on a Dionex UltiMate 3000 RSLC system (Thermo Scientific) followed by mass detection on Orbitrap QExactive (Thermo Fisher Scientific) mass spectrometer (Thermo Fisher). Analysis was operated in polarity switching mode, using 10 μl injection volume and a flow rate of 300 μl/min.

The analyses was performed in four replicates, and a set of standards was run in parallel. Metabolite identification was based on matches with standards where possible or otherwise predicted based on mass and retention time. Data were analysed using mzMatch [31] and mzMatch.ISO [32], Xcalibur software, version 4.0 (Thermo Fisher Scientific, San Jose, CA, USA), and an in-house developed Metabolite Mapper platform. The raw data is publically available from the Figshare depository (https://figshare.com/authors/Julie_Kovarova/17814896).

## FBPase assay

2 x 10$^7$ cells were used per sample. They were centrifuged (1,300 g, 10 min), washed with 1 x PBS and resuspended in 100 μl of SoTE buffer (10 mM Tris-HCl pH 8, 1 mM EDTA, 0.15% Triton X-100, protease inhibitor cocktail (Roche)). After 20 min incubation at RT, samples were centrifuged (14,000 g, 10 min, 16˚ C) and supernatant collected. The reaction mixture containing 20 mM Tris pH 7.8, 10 mM MgCl$_2$, 1 mM NADP, 1 U PGI (Sigma-Aldrich), 1 U G6PDH, 100 μl cell extract was incubated at 30˚ C for 5 min prior to activity measurement. The reaction was triggered by addition of 5 mM F1,6bP (Sigma-Aldrich) immediately prior to measurement of NADPH production at 340 nm for 5 min at 30˚ C using a UV-1601 spectrophotometer (Shimadzu).

## Immunofluorescence assay (IFA)

For the immunofluorescence assay, cells were fixed in 7.4% formaldehyde in 1 x PBS for 15 min, and subsequently washed three times with 1 x PBS. For permeabilisation, 0.1% Triton X-100 (AppliChem) in 1 x PBS was applied for 10 min, and subsequently washed 3 x with 1 x PBS. Following blocking in 5.5% FBS in 1 x PBS-Tween (0.05% Tween) and 2 x wash with 1 x PBS, the primary antibodies were applied (α-Ty at 1:100 (ThermoFisher) and α-FBP at 1:1,000) for 1 h at RT. Following three washes with 1 x PBS-T and two washes with 1 x PBS, the secondary antibodies were applied (Alexa Fluor 647 α-mouse (Life Technologies) at 1:2,000 and Alexa Fluor 488 α-rabbit (Life Technologies) at 1:2,000) for 1 h at RT. After an additional three washes with 1 x PBS-T and two washes with PBS, ProLong Gold Antifade mounting solution (Invitrogen) was applied. Imaging was performed using an Axioplan microscope (Zeiss).

## Results

### Generation of double knock-out *Δfbp.sbp* by two step Cas9 editing

We took advantage of CRISPR/Cas9-based gene editing [27] to generate a double knock-out of FBPase and SBPase (*Δfbp.sbp*) in *T. brucei* BSF. First, an sgRNA template and a repair template were simultaneously electroporated in a transient transfection, resulting in replacement of both alleles of the *FBPase* gene with a G418 resistance cassette. Subsequently, applying the same approach, both alleles of *SBPase* were replaced with a puromycin cassette (Fig 1B). Full

replacement of all alleles was validated by PCR and western blotting (Fig 1C and 1D). Such a straightforward replacement of both target genes indicated that FBPase and SBPase most likely do not play essential roles under the growth conditions used. Accordingly, the *Δfbp.sbp* cell line showed no growth defect in the standard nutrient-rich HMI-11 medium (Fig 1E). When Creek's Minimal Medium (CMM), which better approximates the composition of blood serum, was used for cell culture, *Δfbp.sbp* cells suffered a mild growth defect with higher variability, possibly due to different metabolic adaptations required for growth in CMM (Fig 1E) [33]. Because these experiments showed a growth defect under nutrient-restrictive conditions, we decided to assess infectivity *in vivo*. Infection of mice showed that *Δfbp.sbp* cells were infectious, although they reached lower parasitaemia on day 2 compared to the parental cell line (0.6-fold, p = 0.04 (S1 Fig)).

## Metabolomic analysis of *Δfbp.sbp* shows sugar phosphate substrate accumulation and maintained gluconeogenesis

We next examined the effects of the *fbp* and *sbp* double gene deletion on central carbon metabolism. First, a metabolomic analysis was performed on *Δfbp.sbp* cells cultured in CMM. The most striking change was approx. 30-fold accumulation of S1,7bP (p = 3 x $10^{-6}$), the substrate of SBPase (Fig 2A and 2B), confirming disruption of the SBPase reaction. Sedoheptulose 7-phosphate (S7P) was 4-fold increased (p = $10^{-5}$), probably due to non-enzymatic conversion of highly accumulated S1,7bP. The metabolites of the FBPase reaction were also affected, although to a much lesser extent than for SBPase, with F6P increased 1.3-fold (p = 0.03), and F1,6bP not changed significantly (Fig 2B). Considering that F6P and F1,6bP are also involved

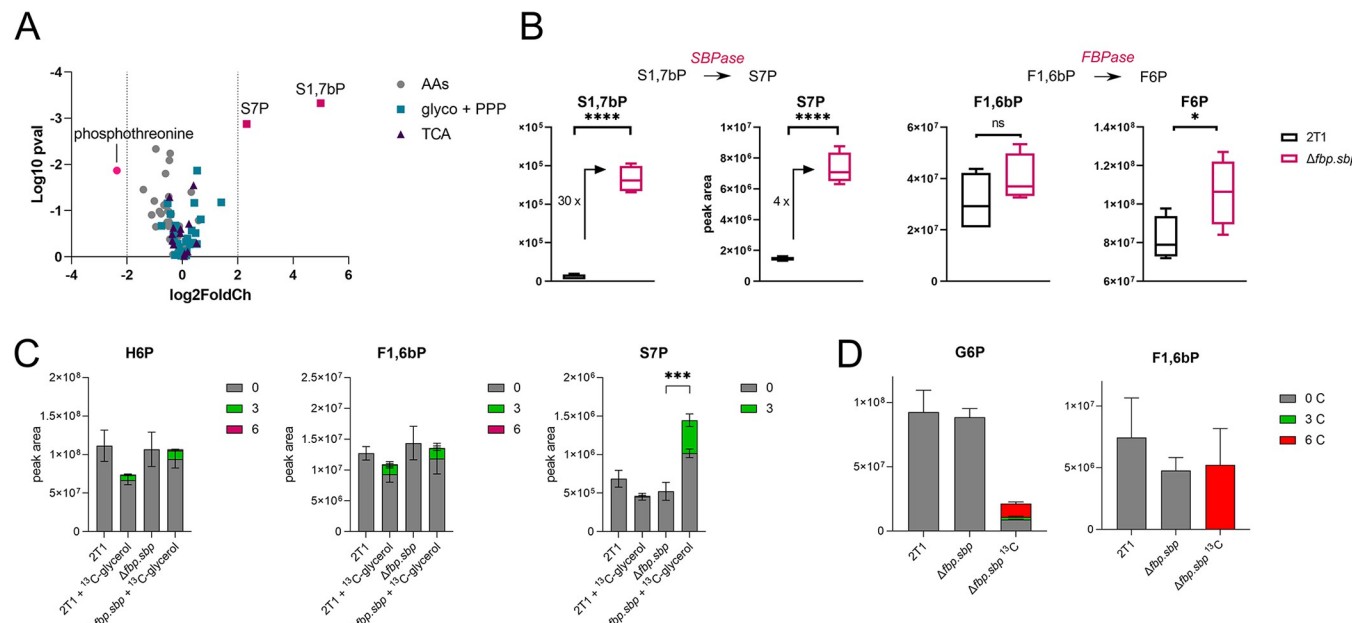

**Fig 2. Gluconeogenic activity is maintained in *Δfbp.sbp* strains.** A–Changes in *Δfbp.sbp* cells grown in CMM and subjected to LC-MS metabolomics when compared to the parental cell line. The most changed metabolites are S7P, S1,7bP, and phosphothreonine. The plot includes 87 metabolites from glycolysis, PPP, TCA cycle or amino acid metabolism. B–Sedoheptulose 1,7-bisphosphate (S1,7bP), sedoheptulose 7-phosphate (S7P), fructose 1,6-bisphosphate (F1,6bP), and fructose 6-phosphate (F6P) as detected in the parental and *Δfbp.sbp* cells grown in CMM and subjected to LC-MS metabolomics. C–*Δfbp.sbp* cells were grown in HMI-11 supplemented with U-$^{13}$C$_3$-glycerol and analysed by LC-MS metabolomics, showing that $^{13}$C from glycerol is incorporated into GNG products. H6P –hexose 6-phosphates. D—*Δfbp.sbp* cells were grown in CMM supplemented with glucose, or U-$^{13}$C$_3$-glycerol and analysed by LC-MS metabolomics. G6P –glucose 6-phosphate.

in other metabolic pathways, particularly glycolysis, the absence of FBPase is not expected to have such a large impact on their relative abundance.

To test whether GNG flux is present, we grew the parental and *Δfbp.sbp* cells in HMI-11 medium supplemented with 5 mM U-$^{13}$C$_3$-glycerol and subjected the samples to metabolomic analysis. Overall, few changes were observed in the *Δfbp.sbp* metabolome compared to the parental cell line, with a 2-fold accumulation of S7P being most evident in *Δfbp.sbp* cells grown with glycerol (Fig 2C). Since glucose was also present in the medium, the predominant portion of the hexose phosphate isotopomer pool was unlabelled in both cell lines, however, $^{13}$C$_3$ labelling represented 10% of the hexose phosphate pool. 12% of F1,6bP contained the $^{13}$C$_3$ label, providing clear evidence that these labelled carbons originate from U-$^{13}$C$_3$-glycerol fed through GNG, in both parental and *Δfbp.sbp* cell lines (Fig 2C). Further, we performed LC-MS metabolomics on *Δfbp.sbp* cells grown in CMM medium supplemented with U-$^{13}$C$_3$-glycerol, which confirmed incorporation of glycerol into G6P and F1,6bP (Fig 2D), and hence presence of GNG. Overall, levels of glycolytic intermediates were decreased when glycerol was the major carbon source compared to glucose (Fig 2D).

Changes in the metabolome of *Δfbp.sbp* cells grown in CMM (Fig 2B) correspond with the growth defect observed in that medium and reflect higher sensitivity of cells to perturbations in nutrient-restricted conditions, in contrast to the HMI-11 medium, where minimal changes in growth and the metabolome of *Δfbp.sbp* cells were observed (Figs 1E and 2C). Overall, the differences detected in sugar phosphates correlate with expectations for FBPase and SBPase deletion, except for maintainance of gluconeogenic flux. This maintained flux was also confirmed by conversion of F1,6bP to F6P in cell extracts; comparable levels of enzymatic activity were measured using a specific enzyme activity assay from the parental cell line (15 μmol/min/mg, n = 3) and *Δfbp.sbp* (16 μmol/min/mg, n = 3).

## Metabolomic data show that depletion of PFK reduces glycolysis but not GNG

Since we did not observe changes in GNG flux in *Δfbp.sbp* cell lines, we next examined whether the reverse activity of PFK (dephosphorylation of F1,6bP into F6P) can contribute to GNG. PFK was depleted by RNAi in the parental cell line (PFK$^{RNAi}$) and in the *Δfbp.sbp* background (*Δfbp.sbp*/PFK$^{RNAi}$). Induction of knockdown using tetracycline caused a severe growth defect in both cell lines, as expected due to the essential role of PFK in glycolysis (Fig 3A). RT-PCR confirmed a drop in target gene expression to 20% in the parental background and 60% in the *Δfbp.sbp* background (insets, Fig 3A). We performed metabolomic analysis of the PFK$^{RNAi}$ and *Δfbp.sbp*/PFK$^{RNAi}$ cells fed with U-$^{13}$C$_3$-glycerol as the sole carbon source (no glucose was present in the HMI-11 medium, except for ~0.5 mM glucose from 10% FBS). Samples were harvested 24 h after tetracycline induction in the glycerol medium and subjected to LC-MS metabolomics. The profile of glycerol 3-phosphate (G3P) shows that glycerol was taken up and incorporated, since the $^{13}$C$_3$-labelled part represents around 80% (78–85%) of the total in all sample groups (Fig 3B). Crucially, RNAi induced cells with decreased PFK expression still utilised glycerol via GNG and incorporated it into hexose phosphate sugars. The profile of F6P shows that the majority (44–66%) is $^{13}$C$_6$-fructose 6-phosphate, made from $^{13}$C$_3$-glycerol, in all samples (Fig 3B). The major difference after PFK depletion is in the unlabelled part that originates from glucose via glycolysis, and relies on the forward reaction of PFK. This fraction decreased from 18 to 8% of the total amount in PFK$^{RNAi}$ induced samples, confirming a reduction in PFK glycolytic activity. On the other hand, the proportion of the $^{13}$C$_6$-labelled part increased from 44 to 66%, demonstrating undiminished glycerol incorporation. A substantial part of F6P is $^{13}$C$_3$-labelled (26–38%), indicating that both reactions,

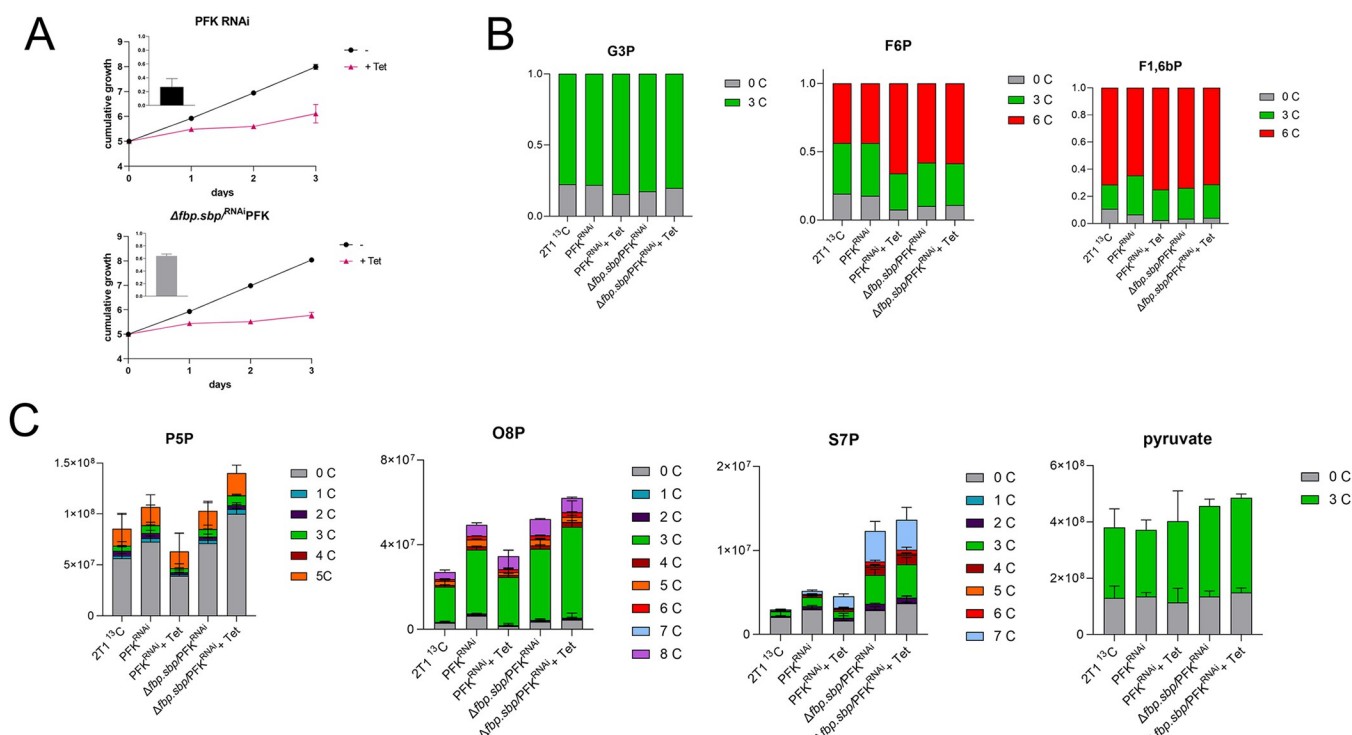

**Fig 3. PFK knockdown causes a severe growth defect, but only mild changes in the metabolome.** A–Growth curves of PFK^RNAi and in *Δfbp.sbp*/^RNAiPFK show a severe growth defect after RNAi. Levels of PFK mRNA are shown as detected by qRT-PCR, compared to non-induced cells, and normalised to 18S, 24 h tetracycline induction (insets). B–LC-MS metabolomics in HMI-11 medium depleted of glucose, and supplemented with $^{13}C_3$-glycerol. The proportion of the $C_{13}$ label incorporation relative to the total amount is indicated. G3P –glycerol 3-phosphate, F6P –fructose 6-phosphate, F1,6bP–fructose 1,6-bisphosphate. 2T1 $^{13}C$ –parental cell line in medium with $^{13}C$-glycerol, PFK^RNAi–non-induced PFK RNAi cell lines in medium with $^{13}C$-glycerol, PFK^RNAi + Tet–PFK RNAi induced for 24 h in medium with $^{13}C$-glycerol, *Δfbp.sbp*/PFK–non-induced *Δfbp.sbp*/^RNAiPFK cell line in medium with $^{13}C$-glycerol, *Δfbp.sbp*/PFK + Tet— *Δfbp.sbp*/^RNAiPFK cell line induced for 24 h in medium with $^{13}C$-glycerol. C–Same as in B), but relative changes in the total levels of metabolites are depicted. P5P –pentose 5-phosphates, O8P –octulose 8-phosphate, S7P –sedoheptulose 7-phosphate.

phosphorylation of F6P into F1,6P and the reverse dephosphorylation, take place. Analysis of the F1,6bP profile shows an even higher proportion of $^{13}C_6$-labelled, representing 64–74% across the samples. Smaller changes occur after induction of PFK knockdown, but the same trend as in F6P is observed. The unlabelled part dropped from 6.5 to 2.5%, $^{13}C_3$ decreased from 29 to 22%, and $^{13}C_6$ increased from 65 to 75% (Fig 3B). In the *Δfbp.sbp*/^RNAiPFK cell line, no significant changes were detected after induction of PFK silencing, suggesting that any further perturbation of metabolic flux is lethal for these cells.

Pentose phosphate pathway (PPP) intermediates are labelled to a lesser extent than the hexose phosphates. Around 70% of the total pentose phosphate pool is unlabelled, while $^{13}C_5$ represents between 11 to 18%, and $^{13}C_3$ about 5% (Fig 3C). Octulose 8-phosphate (O8P) has a similar labelling pattern, suggesting its synthesis from pentose phosphates condensed with a three carbon moiety ($^{13}C_3$ represents ~ 65%, Fig 3C). S7P, another intermediate of the non-oxidative PPP, is substantially labelled. The $^{13}C_3$ part comprised around 20% of total S7P in the WT control and PFK^RNAi, and 28% in *Δfbp.sbp*/^RNAiPFK cells, regardless of whether knockdown was induced or not, suggesting that PFK depletion has little impact on the incorporation of glycerol. However, depletion of FBPase and SBPase caused an increase in $^{13}C_3$, in addition to the increase in total amount of S7P more than 4-fold (compared to WT) (Fig 3C), consistent with the previous measurement (Fig 2B). Moreover, the $^{13}C_7$ part represented about 30% of total S7P in *Δfbp.sbp* cells, confirming glycerol as a substrate for the production of S7P.

These larger sugar phosphates have been observed in *T. brucei* before [29], and although their role in metabolism remains unclear, the current metabolomic data offers new insights into their biosynthesis.

Metabolomic profiles of other metabolites, such as intermediates of glycolysis, the succinate shunt, or mitochondrial carbon metabolism, provide additional valuable information. Substantial $^{13}C_3$ labelling was observed in phosphoenolpyruvate (PEP), pyruvate, fumarate, malate, aspartate, and alanine (Figs 3C and S2). 72–78% of PEP is $^{13}C_3$, demonstrating a high level of utilisation of labelled glycerol in glycolysis. The unlabelled fraction dropped from 22% to 13% after PFK knockdown, again confirming a drop in glucose utilisation in glycolysis, relative to glycerol. Pyruvate comprised 28–36% of unlabelled, and 60–68% of $^{13}C_3$-pyruvate. Malate comprised 43% to 57% of unlabelled, 10–16% of $^{13}C_2$ (synthesized in the mitochondrion via acetyl-CoA), and 28% - 36% of $^{13}C_3$ (synthesized in the glycosomal succinate shunt), without significant differences between the sample groups (S2 Fig). For aspartate, the $^{13}C_3$ content ranges from 42% to 50% in all cell samples. These data indicated that glycerol was used in glycolysis for pyruvate and acetyl-CoA production and further fed into mitochondrial enzyme reactions classically associated with the TCA cycle (S2 Fig). The data are consistent with our previous work, where we observed similar labelling patterns with glycerol, indicating an increase in mitochondrial metabolism when BSF cells rely on GNG [6]. Overall, few changes were observed after PFK knockdown, despite the severe growth defect. However, we saw clear evidence for diminished glycolysis, but not GNG. The data demonstrated that glycerol was utilised as the main carbon source under the experimental conditions, since the majority of glycolytic intermediates were labelled from $^{13}C_3$-glycerol. In addition to use as a substrate for GNG, glycerol was used for synthesis of S7P, which may reflect a novel variation of the non-oxidative PPP.

## PFK and FBPase co-localise in glycosomes

We next sought to establish the localisation of the key glycolytic and GNG enzymes, PFK and FBPase. Both of these enzymes are glycosomal, but their mutual co-localisation has never been reported. For this purpose, one of the endogenous PFK alleles was tagged in the 2T1$^{T7-Cas9}$ cell line with the Ty-tag, generating $^{Ty}$PFK, which was used for microscopy in a combination with a specific α-FBPase antibody. We cultured the cells in medium supplemented with glucose or glycerol only, or in a combination of both carbon sources, and then subjected these cells to super-resolution confocal microscopy (Fig 4). PFK and FBPase always co-localised in glycosomes and no differences were observed under the various conditions tested. The two enzymes are not separated by compartmentalisation. Further work will be required to reveal potential regulation, although futile cycling does occur to some extent as seen in the metabolomic data (Fig 3B).

## TAL does not contribute to gluconeogenesis

One possible explanation for the observed GNG activity (i.e. generation of F6P from glycerol) may be a novel metabolic pathway involving three enzymes (SBPase, TAL, and fructose-1,6-bisphosphate aldolase) that operate in a cycle, using glycolytic intermediates as substrates to produce F6P (Fig 5A) [25]. Considering the unprecedented presence of SBPase, expression of TAL in the absence of transketolase, and the persistence of GNG flux, such a novel pathway could offer an explanation. Hence, we decided to test the role of TAL in GNG, and generated a knock-out cell line lacking *FBPase*, *SBPase*, and *TAL* (*Δfbp.sbp.tal*). *TAL* was deleted using Cas9 and replaced with a phleomycin resistance cassette in the parental 2T1$^{T7-Cas9}$ and in the *Δfbp.sbp* cell lines (Fig 5B). The *Δfbp.sbp.tal* cells displayed no growth defect in HMI-11

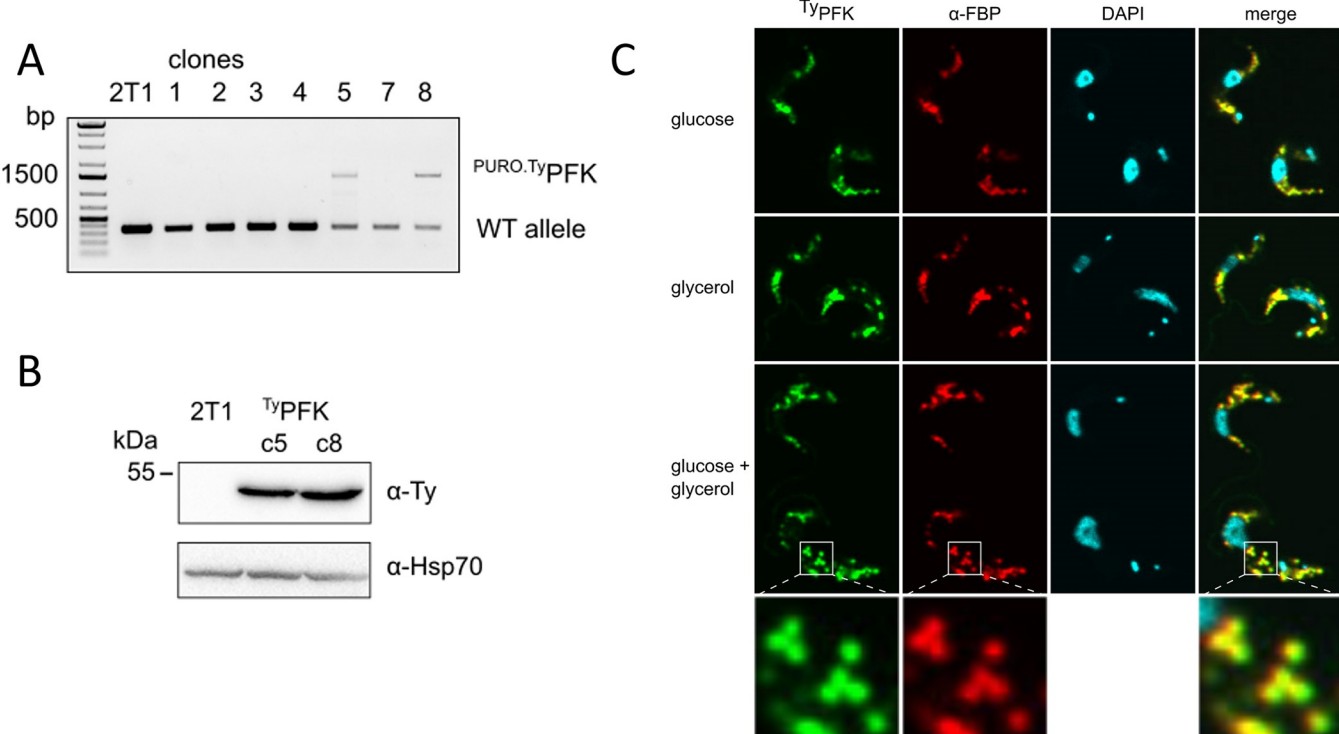

**Fig 4. PFK and FBPase co-localise in glycosomes.** A–PCR validation of ^Ty^PFK clones. B–Western blot validation of the ^Ty^PFK cell line. C–Immunofluorescence assay with ^Ty^PFK cell line, α-Ty, and α-FBPase staining. Cells were cultured in CMM with different carbon sources (glucose, glycerol, or both), but it had no effect on co-localisation of PFK and FBPase.

medium supplemented either with glucose or glycerol as the main carbon source (Fig 5C). The cells were also still infective to mice at levels comparable to the parental cell line (S3 Fig).

LC-MS metabolomic analysis was performed on *Δfbp.sbp.tal* cells grown in CMM supplemented with 5 mM $^{13}C_3$-glycerol and no glucose (~0.5 mM glucose was present due to supplementation with 10% FCS). The results showed substantial uptake of, and reliance upon, glycerol as the major carbon source. The profile of G6P is surprisingly similar between the parental and *Δfbp.sbp.tal* cells when fed with $^{13}C_3$-glycerol, with the majority of the G6P metabolite being $^{13}C_6$-labelled (62% in WT and 49% in *Δfbp.sbp.tal*) (Fig 5D). Even more substantial is the fully labelled part in F1,6bP, representing 84% of total in WT and 78% in *Δfbp.sbp.tal*. Additionally, the total amount of the G6P metabolite is 2-fold increased in the triple knock-out (Fig 5D). S7P is a sugar phosphate formed by an unknown mechanism, however, our data show that S7P is synthesized from glycerol. The total amount of S7P is increased 3-fold in WT when fed on glycerol relative to glucose, indicating that glycerol increases S7P synthesis. In addition, S7P is increased 2-fold in *Δtal* and 6-fold in the *Δfbp.sbp.tal* cell line. The majority is $^{13}C_3$- and $^{13}C_7$-labelled (28% and 49% respectively, in *Δfbp.sbp.tal*, Fig 5D). In summary, our metabolomic data show persistence of GNG flux from glycerol even in the *Δfbp.sbp.tal* cell line. We did not observe other major changes in the metabolome associated with deletion of TAL, thus the function of TAL in BSF trypanosomes, where its usual PPP partner enzyme transketolase is absent, remains unknown.

## Discussion

We took advantage of Cas9-based gene editing to delete multiple genes and in an attempt to abolish GNG in BSF *T. brucei*. However, our data show that the combined deletion of FBPase,

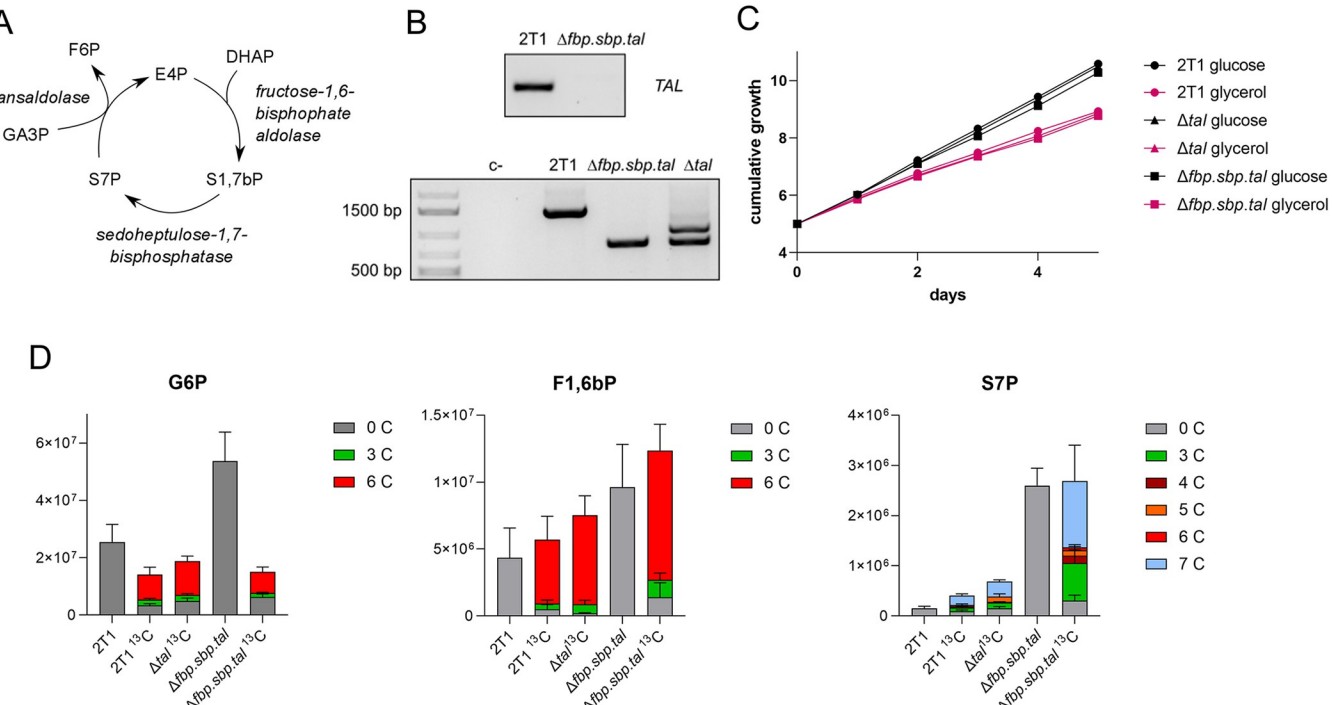

**Fig 5. Deletion of TAL does not decrease GNG flux.** A—The novel pathway as suggested by Hannaert [25]: Erythose 4-phosphate (E4P) is condensated with dihydroxyacetone phosphate (DHAP) by fructose-1,6-bisphosphate aldolase into S1,7bP, which is dephosphorylated by SBPase into S7P. That is used as a substrate by transaldolase, together with glyceraldehyde 3-phosphate (GA3P), and converted into F6P and E4P, which can enter another cycle. B—PCR detection of a 500-bp product from the *TAL* ORF, proving deletion of the gene (top panel). PCR with oligos annealing to to the UTRs of the *TAL* gene, amplifying the *TAL* gene in the parental cell line (1,402 bp), replaced by phleomycin resistance gene (775 bp) in *Δfbp.sbp.tal*, and both phleomycin resistance and puromycin N-acetyltransferase (1,006 bp) in *Δtal* (bottom panel). C—Growth curves of the parental (2T1^T7-Cas9), *Δtal*, and *Δfbp.sbp.tal* cell lines in HMI-11 medium supplemented with glucose, or glycerol only (and 10% FBS). D—LC-MS metabolomics in CMM supplemented with 5 mM $^{13}C_3$-glycerol. G6P – glucose 6-phosphate, F1,6bP—fructose 1,6-bisphosphate, S7P - sedoheptulose 7-phosphate. 2T1 –parental cell line in medium with glucose, 2T1 $^{13}$C –parental cell line in medium with $^{13}$C-glycerol, *Δtal* $^{13}$C –*Δtal* cell line in medium with $^{13}$C-glycerol, *Δfbp.sbp.tal*—*Δfbp.sbp.tal* cell line in medium with glucose, *Δfbp.sbp. tal* $^{13}$C - *Δfbp.sbp.tal* cell line in medium with $^{13}$C-glycerol.

SBPase, and TAL had little impact on gluconeogenic activity. To the best of our knowledge, this is the first case of deletion of three diploid gene loci simultaneously in BSF *T. brucei*. Although depletion of PFK was possible only to a limited extent, our data show that the reverse activity of this enzyme is most likely not involved in GNG, as suggested previously [17]. Current understanding of trypanosome metabolism does not provide an alternative explanation for how GNG operates, and which enzymes contribute to GNG flux.

In our previous work and that of others, it was demonstrated that deletion of the canonical *FBPase* gene in *T. brucei* does not deplete FBPase activity, i.e. conversion of F1,6bP to F6P [6,7]. This is in contrast to closely related *Leishmania* parasites, where *FBPase* deletion disrupted GNG and caused a severe phenotype in mammalian infective amastigotes [34]. Since SBPase is not present in the genome of *Leishmania*, this enzyme was a promising candidate for the observed FBPase activity. The enzyme has 26% sequence identity to FBPase, and its catalytic activity is predicted to be very similar, using a sugar phosphate backbone extended by one carbon. Additionally, SBPase from yeast was demonstrated to posess 'FBPase' activity [35]. The most probable origin of SBPase in trypanosomes is acquisition by horizontal gene transfer [36].

The streamlined protocol for Cas9-based gene editing allowed us to generate a double gene knock-out combined with a knock-down. Knock-out for both *FBPase* and *SBPase* genes were readily obtained by sequential transfections. The *Δfbp.sbp* cell line had no growth defect under

nutrient-rich culture conditions. However, there was a mild and variable growth defect in CMM. The high variability in growth rate could be caused by different mechanisms of adaptation to nutrient-restricted conditions. The metabolomic experiments with *Δfbp.sbp* revealed that SBPase bears its canonical enzymatic activity (converting S1,7bP to S7P), since its substrate, S1,7bP was highly accumulated in the knockout cell line. A concomitant increase in S7P is most likely explained by non-enzymatic loss of a single phosphate from the accumulated S1,7bP. Similarly, accumulation of S1P and S1,7bP was observed previously in an SBPase yeast deletion mutant [37]. SBPase was also identified in *Toxoplasma gondii*, and its deletion resulted in a similar phenotype to what we observed here, both in metabolomics and in decreased infectivity [38].

Fernandes and colleagues [17] showed reverse PFK activity (equal to canonical FBPase) *in vitro* and proposed that the glycosomal microenvironment might create conditions for reverse PFK activity. However, our data do not support this view because, although depletion of PFK by RNAi was not complete (decrease in PFK expression to 20% in parental and to 60% in the *Δfbp.sbp* background), we could detect a major decrease in glycolytic flux but not in GNG flux. However, if reactions in both directions occur simultaneously, we would not be able to detect a minor change in GNG flux. More efficient PFK knockdown by RNAi is unlikely to be achievable in the *Δfbp.sbp* double knock-out mutant, because the metabolic flux is flexible and the enzymes can compensate for each other to some extent, which is impossible in the double knock-out background. Metabolomics showed surprisingly few changes in the levels of detected metabolites in the PFK knockdown cell lines. Nevertheless, we noted a decrease in glycolytic flux, but not in GNG flux. The directionality of the PFK activity is dependent on concentrations of substrates and products and the negative free energy ($\Delta$G) of the PFK reaction is highly favorable for the forward reaction. Significant perturbations in substrate concentrations including ADP/ATP ratio and changes in glycosomal pH might allow the PFK reversal [17]. Nevertheless, in our metabolomic datasets we did not detect any prominent changes in ADP, ATP, F6P, and F1,6bP levels, for instance one of the largest changes is the 2-fold accumulation of F1,6bP in *Δfbp.sbp*/[RNAi]PFK compared with WT (S3 Fig). Altogether, our data do not support participation of PFK in GNG. However, it should be noted that we measured metabolites extracted from whole cells, whereas metabolite levels and especially ADP/ATP ratios may be significantly shifted within the glycosomal subcompartment compared to that of the whole cell. The application of technically challenging methods is required to determine the metabolic microenvironment of glycosomes to gain a deeper understanding of the involvement of individual enzymes in GNG.

The PPP intermediates detected by metabolomics can be separated into two groups. The first represents the oxidative PPP and metabolites derived originally from glucose, where less labelling from glycerol was detected (pentose phosphates, Fig 3C). The second group, representing the non-oxidative PPP, contains S7P or O8P, which have significant proportions labelled (Figs 3C and 5D). Since BSF *T. brucei* does not express transketolase [20,21], the metabolites cannot be produced in the canonical non-oxidative PPP. This suggests that in a low concentration of glucose, as used here, glucose is used preferentially to feed the oxidative PPP, whereas glycerol is used to feed glycolysis and to synthesize large sugar phosphates such as S7P. Although O8P has previously been shown to be synthesized *in vitro* from F6P and R5P by TAL [29], there appears to be a separate route for synthesis in the parasites, as evidenced by the presence of this metabolite in *TAL*-depleted cell lines. In both WT and *Δfbp.sbp.tal*, much higher quantities of O8P were detected in the presence of glucose than in the same cell lines grown in glycerol-based medium (S4 Fig). Our datasets provide new insights into the role of TAL and the whole PPP in BSF, however, further studies will be required to resolve the arrangement of the pathway in this life stage.

A novel metabolic pathway that includes the activities of SBPase, TAL, and fructose-1,6-bisphosphate aldolase was proposed [25]. If active, this pathway conforms to the labelling patterns detected in S7P (and O8P), and also with S1,7bP production catalysed by aldolase, as reported previously [37]. Most importantly, this would present an alternative pathway for the production of F6P from glycerol. Nevertheless, our experiments with the *Δtal* and *Δfbp.sbp.tal* mutants show that carbons from glycerol are still incorporated into F1,6bP, F6P, and other metabolites, demonstrating continued flux though GNG, and precluding the existence of the alternative pathway.

It is also unclear how the flux between glycolysis and GNG, i.e. between PFK and FBPase activity, is regulated. Potentially, the forward or reverse flux could be controlled by upstream kinases, i.e. hexokinase in glycolysis and glycerol kinase in GNG, due to competition for ATP, as reported recently [39]. Another possibility would be an exclusive compartmentalisation of these enzymes, but our immunofluorescence analysis shows that PFK and FBPase co-localise under all conditions tested. These results suggest that some futile cycling does take place, as substantial $^{13}C_3$-labelling of hexose phosphates (Fig 3B) occurs. However, since FBPase is not a key GNG enzyme, we suggest that regulation is achieved by a distinct enzyme that possesses FBPase activity but whose identity is yet to be established.

When glucose is limiting, bloodstream form trypanosomes adapt by activating alternative metabolic pathways. Labelling patterns in succinate and citrate indicate flux in mitochondrial metabolism. Since only a small proportion of labelling in citrate was detected (72–89% of citrate was unlabelled), incorporation of additional substrates, such as glutamine or other amino acids likely occurs, as reported previously [40]. If glycerol is available to trypanosomes, it is used as a carbon source, being incorporated into sugar phosphates under all of conditions tested here. Deletion or depletion of four different enzymes failed to substantially diminish GNG flux in BSF trypanosomes. Taken together, our results indicate that (an)other enzyme(s), currently unrecognized by *in silico* searchers, is/are responsible for the activity. The question as to whether GNG is essential for BSF *T. brucei* cannot be readily addressed until these key enzyme(s) are identified.

## Supporting information

**S1 Fig. Mice infection with *Δfbp.sbp* cell line.**
(TIF)

**S2 Fig. LC-MS metabolomics in HMI-11 medium depleted of glucose, but supplemented with $^{13}C_3$-glycerol.** 2T1 $^{13}C$ –parental cell line in medium with $^{13}C$-glycerol, PFK$^{RNAi}$–non-induced PFK RNAi cell lines in medium with $^{13}C$-glycerol, PFK$^{RNAi}$ + Tet–PFK RNAi induced for 24 h in medium with $^{13}C$-glycerol, *Δfbp.sbp*/PFK–non-induced *Δfbp.sbp*/$^{RNAi}$PFK cell line in medium with $^{13}C$-glycerol, *Δfbp.sbp*/PFK + Tet—*Δfbp.sbp*/$^{RNAi}$PFK cell line induced for 24 h in medium with $^{13}C$-glycerol.
(TIF)

**S3 Fig. Mice infection with *Δfbp.sbp.tal* cell line.**
(TIF)

**S4 Fig. Octulose 8-phosphate (O8P) as detected in the *Δtal* cell lines by LC-MS metabolomics.** $^{13}C$ indicates medium supplemented with $^{13}C$-glycerol as the only carbon source.
(TIF)

**S1 Table. Standards for LC/MS metabolomics.**
(XLSX)

## Acknowledgments

We thank Prof. Frederic Bringaud (University of Bordeaux) and Prof. Paul Michels (University of Edinburgh) for the kind gift of antibodies α-FBPase, α-SBPase, and α-PFK.

## Author Contributions

**Data curation:** Julie Kovářová, Martin Moos.

**Formal analysis:** Julie Kovářová, Martin Moos.

**Funding acquisition:** Julie Kovářová, Michael P. Barrett, David Horn, Alena Zíková.

**Investigation:** Julie Kovářová, Martin Moos.

**Methodology:** Julie Kovářová, Martin Moos, Michael P. Barrett.

**Project administration:** Julie Kovářová.

**Resources:** David Horn, Alena Zíková.

**Supervision:** Michael P. Barrett, David Horn.

**Validation:** Julie Kovářová.

**Visualization:** Julie Kovářová.

**Writing – original draft:** Julie Kovářová, Michael P. Barrett, David Horn, Alena Zíková.

**Writing – review & editing:** Julie Kovářová, Alena Zíková.

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
