## [Decision Letter · Decision Letter 0]

21 Nov 2023

Dear Dr. Kovářová,

Thank you very much for submitting your manuscript "The bloodstream form of Trypanosoma brucei displays non-canonical gluconeogenesis" for consideration at PLOS Neglected Tropical Diseases. As with all papers reviewed by the journal, your manuscript was reviewed by members of the editorial board and by several independent reviewers. In light of the reviews (below this email), we would like to invite the resubmission of a significantly-revised version that takes into account the reviewers' comments. 

We cannot make any decision about publication until we have seen the revised manuscript and your response to the reviewers' comments. Your revised manuscript is also likely to be sent to reviewers for further evaluation.

Sincerely,

Walderez O. Dutra, PhD.

Section Editor

Walderez Dutra

Section Editor

Reviewer's Responses to Questions

**Key Review Criteria Required for Acceptance?**

**Methods**

-Are the objectives of the study clearly articulated with a clear testable hypothesis stated?

-Is the study design appropriate to address the stated objectives?

-Is the population clearly described and appropriate for the hypothesis being tested?

-Is the sample size sufficient to ensure adequate power to address the hypothesis being tested?

-Were correct statistical analysis used to support conclusions?

-Are there concerns about ethical or regulatory requirements being met?

Reviewer #1: (No Response)

Reviewer #2: This is a relevant MS raising an important question on Trypanosoma brucei biology: are BSF able to perform glyconeogenesis? The MS herein brings relevant contributions to this issue. The paper is very well conducted and written.

**Results**

-Does the analysis presented match the analysis plan?

-Are the results clearly and completely presented?

-Are the figures (Tables, Images) of sufficient quality for clarity?

Reviewer #1: (No Response)

Reviewer #2: Results are solid and based in well designed and sophisticated exeriments.

**Conclusions**

-Are the conclusions supported by the data presented?

-Are the limitations of analysis clearly described?

-Do the authors discuss how these data can be helpful to advance our understanding of the topic under study?

-Is public health relevance addressed?

Reviewer #1: (No Response)

Reviewer #2: Conclussions are in agreement with the content of the MS.

**Editorial and Data Presentation Modifications?**

Reviewer #1: (No Response)

Reviewer #2: Not needed.

**Summary and General Comments**

Reviewer #1: In this manuscript, Kovářová et al investigate the enzymes involved in gluconeogenesis in bloodstream form T. brucei. Overall, this is a well-designed study with some intriguing findings. However, I have the following concerns:

Major concerns:

1. Parameters for metabolomics data processing should be provided, with particular emphasis on methods used to provide and confirm metabolite annotations.

2. Multiple lines in the manuscript refer to targeted metabolomics (eg 403, 491), yet the methods section only describes untargeted metabolomics approaches. If targeted approaches were performed (for example, data acquisition in PRM mode), then the inclusion list should be provided.

3. Given the surprising findings of lack of effect of the KO on 13C incorporation, data supporting the annotations of the metabolites displayed in Fig 3, Fig 5, Fig S2 should be provided, such as retention time matching to the standards run by the authors.

4. The growth defect of the Δfbp.sbp is most pronounced in CMM media, and the untargeted metabolomic changes reported in Fig 2AB are likewise from growth in CMM media. It is therefore critical that the 13C tracing experiments for the double KO strain also be performed in CMM media. That would also enable comparison with the triple KO in Figure 5D.

5. S1 figure: if available, adding additional data covering multiple acute-stage timepoints or additional replicates would provide stronger confirmation that the KO strains do indeed present with a growth defect. The fact that the triple KO strain in Figure S3 showed no phenotype suggests the possibility that the findings in Figure S1 might not be reproducible (especially since the magnitude of the phenotype was so small).

6. Based on the primers provided in Table 1, the product confirming SBP deletion should be 612 bp, but in Fig 1C, it’s marked at 700 bp. The full gel with full ladder should be provided for both panels in Fig 1C, to confirm that products are at the expected size.

7. Table 1 indicates that primers were also used to confirm replacement with the drug resistance cassette (not just loss of the internal primer amplification product). That data is not provided in the manuscript but should be added.

Minor concerns:

1. I would recommend that authors deposit their metabolomics data in a public data repository.

2. Clone numbers should be added to Figure 4A, so that readers can confirm that the clones with the correct Western blot band in Figure 4B match with the clones with the correct PCR band in Figure 4A.

3. Did the triple KO strain show a growth defect in CMM media? Such data would provide an interesting complement to the in vivo data and to the growth data for the double KO, but is not mandatory.

4. Typo line 539, leishmania should be capitalized and italicized.

Reviewer #2: As mentioned above, this MS approaches a relevant question to the trypanosomatids community. The manuscript looks carefully made and written, bringing invaluable information to the field. I made actually a small number of minor recomendations in the spirit of contributing with the improvement of the manuscript.

PLOS authors have the option to publish the peer review history of their article (what does this mean?). If published, this will include your full peer review and any attached files.

Reviewer #1: No

Reviewer #2: No
---

## [Decision Letter · Decision Letter 1]

16 Feb 2024

Dear Dr. Kovářová,

We are pleased to inform you that your manuscript 'The bloodstream form of Trypanosoma brucei displays non-canonical gluconeogenesis' has been provisionally accepted for publication in PLOS Neglected Tropical Diseases.

Best regards,

Walderez O. Dutra, PhD.

Section Editor

Walderez Dutra

Section Editor

Reviewer's Responses to Questions

**Key Review Criteria Required for Acceptance?**

**Methods**

-Are the objectives of the study clearly articulated with a clear testable hypothesis stated?

-Is the study design appropriate to address the stated objectives?

-Is the population clearly described and appropriate for the hypothesis being tested?

-Is the sample size sufficient to ensure adequate power to address the hypothesis being tested?

-Were correct statistical analysis used to support conclusions?

-Are there concerns about ethical or regulatory requirements being met?

Reviewer #1: (No Response)

Reviewer #2: (No Response)

**Results**

-Does the analysis presented match the analysis plan?

-Are the results clearly and completely presented?

-Are the figures (Tables, Images) of sufficient quality for clarity?

Reviewer #1: (No Response)

Reviewer #2: (No Response)

**Conclusions**

-Are the conclusions supported by the data presented?

-Are the limitations of analysis clearly described?

-Do the authors discuss how these data can be helpful to advance our understanding of the topic under study?

-Is public health relevance addressed?

Reviewer #1: (No Response)

Reviewer #2: (No Response)

**Editorial and Data Presentation Modifications?**

Reviewer #1: (No Response)

Reviewer #2: (No Response)

**Summary and General Comments**

Reviewer #1: All my comments have been satisfactorily addressed.

Reviewer #2: The authors have discussed approppriately my comments. I hope they found them useful.

PLOS authors have the option to publish the peer review history of their article (what does this mean?). If published, this will include your full peer review and any attached files.

Reviewer #1: No

Reviewer #2: **Yes: **Ariel Mariano Silber

---

## [Editor Report · Acceptance letter]

19 Feb 2024

Dear Dr. Kovářová,

We are delighted to inform you that your manuscript, "The bloodstream form of Trypanosoma brucei displays non-canonical gluconeogenesis," has been formally accepted for publication in PLOS Neglected Tropical Diseases.

Best regards,

Shaden Kamhawi

co-Editor-in-Chief

Paul Brindley

co-Editor-in-Chief
